# Associations between Duration of Homelessness and Cardiovascular Risk Factors: A Pilot Study

**DOI:** 10.3390/ijerph192214698

**Published:** 2022-11-09

**Authors:** Jie Gao, Haiyan Qu, Keith M. McGregor, Amrej Singh Yadav, Hon K. Yuen

**Affiliations:** 1Department of Clinical and Diagnostic Science, School of Health Professions, University of Alabama at Birmingham, Birmingham, AL 35294, USA; 2Department of Health Service Administration, School of Health Professions, University of Alabama at Birmingham, Birmingham, AL 35294, USA; 3Department of Occupational Therapy, School of Health Professions, University of Alabama at Birmingham, Birmingham, AL 35294, USA

**Keywords:** cardiovascular disease, cardiovascular disease risk factors, homelessness, duration of homelessness

## Abstract

Cardiovascular disease (CVD) in the United States disproportionally affects people who are homeless. This disparity is a critical concern that needs to be addressed to improve the health of individuals who are homeless. The connections between a history of homelessness, i.e., its duration and frequency, and CVD risk are not well understood. The present study sought to investigate how a history of homelessness is correlated with CVD risk factors in a sample of homeless persons in the Deep South. This study recruited participants who were homeless from two local adult homeless shelters in Birmingham, AL. Participants (*n* = 61) underwent interviews, physical measurements, and a capillary blood draw. Their mean age was 47 years, and 82% were men. Results showed the duration of homelessness was positively associated with several CVD risk factors (diabetes mellitus, total cholesterol, and low-density lipoprotein). However, there was no significant association between frequency of homelessness and any CVD risk factors. To get the more accurate estimate of CVD risk in this population, future research should incorporate additional risk factors related to homelessness and seek to develop a robust strategy to collect an accurate history of homelessness.

## 1. Introduction

Studies have shown that cardiovascular disease (CVD) is more prevalent in the homeless population relative to the general population [1,2]. This is likely due to the extra factors created by homelessness that increase challenges in preventing, diagnosing, and managing CVD conditions. People who are homeless often experience long-term food insecurity, poor housing stability and quality, heavy smoking, and excessive substance misuse, all of which are known to be linked to an increased likelihood of CVD [3,4,5,6,7,8]. It is also commonly agreed that many cardiovascular comorbidities such as diabetes mellitus and hypertension are inadequately diagnosed and controlled in people who are homeless due to their limited accesses to healthcare services–a circumstance that often leads to severe complications of CVD and high mortality [9,10,11]. Given the growing homelessness in the United States, the high incidence of CVD in the homeless population has become a public health issue [12].

It is important to note that the homeless population shifts constantly, and most homeless episodes are time-varying transitional periods [13,14]. Therefore, a person’s history of homelessness, such as its duration (i.e., period prevalence) or frequency (i.e., point-in-time counts), can also be a key factor in the development of illness and disease, including CVD. However, little attention has been paid to the effects of a person’s history of homelessness, i.e., its duration and frequency, on health outcomes. Currently, there is only a small body of research about the influences of the duration of homelessness on mental health and sexual health [15,16,17,18,19]. Findings from these studies indicate that a longer time spent homeless generally exacerbates health concerns [15,16,17,18,19]. 

Despite available data, the connections between a history of homelessness and CVD risk factors are not well understood. There remains a need to fully characterize the impacts of homelessness-related factors on modifiable CVD risk factors, which include high levels of total cholesterol and low-density lipoprotein (LDL), low levels of high-density lipoprotein (HDL), and diabetes mellitus. Our previous study [20] evaluated the racial differences in the prevalence of CVD risk biomarkers (i.e., total cholesterol, LDL, HDL, and hemoglobin A1c [HbA1c]) among homeless men, and we are unaware of any studies that have previously examined the relationship between a history of homelessness and these risk factors. Therefore, this study sought to investigate how these CVD risk biomarkers are correlated with a history of homelessness in homeless adults in Birmingham, AL, USA, a state in the Deep South where the prevalence of CVD is disproportionately higher than in other U.S. states.

## 2. Materials and Methods

### 2.1. Participants

A total of 63 participants were recruited at two local shelters for homeless individuals in Birmingham, AL, between July 2021 and April 2022. Participants eligible for this study were adults (aged > 18 years) who met one of the four broad categories of homelessness as defined by the U.S. Department of Housing and Urban Development [21]. Exclusion criteria were the inability to provide informed consent, engage in a daily conversation for 30 min, or give information about the history of homelessness, or being unwilling to donate a sample of capillary blood. Overall, 61 homeless individuals participated in this study, which had an enrollment rate of 96.8%. 

### 2.2. Data Collection

This study used a cross-sectional research design. After obtaining informed consent, investigators interviewed participants and recorded their responses in an online questionnaire created with Qualtrics^XM^ that collected their sociodemographic data, history of homelessness (e.g., frequency and duration of homelessness), alcohol and drug problems, smoking history, and assessed their diet. The sociodemographic information collected included age, gender, race, educational level, marital status, employment status, veteran status, LGBTQ status, disability status, and adverse childhood experiences (present or absent). The self-reported history of homelessness included the duration of participants’ current homeless episode, their number of lifetime homeless episodes (frequency of homelessness), and their usual sleeping place during homelessness.

### 2.3. Description of Instruments

An 8-item dietary screening instrument (Starting the Conversation) [22] was included in the questionnaire to measure the frequency of different food consumption behaviors. Alcohol and drug problems were assessed individually, with four questions each (D19, D23, D26, and D28 for alcohol problems and D20, D24, D27, and D29 for drug problems) from the section “Alcohol/Drugs” in the Addiction Severity Index [23]. These questions collect participants’ experiences with alcohol or drug abuse in their lifetime and in the past 30 days. Responses to questions related to alcohol and drug problems were coded in 4- or 5-point scales. Summary scores ranging from 0 to 17 for alcohol or drug problems were formed by summing the points from each set of four questions, with higher scores indicating a greater number of addictive behaviors. The internal consistency reliability of the alcohol and drug problem variables estimated by Cronbach’s alpha were 0.76 and 0.72, respectively. Smoking history was categorized as never smoker, former smoker, someday smoker, and every day smoker based on the glossary of the National Health Interview Survey from the Centers for Disease Control and Prevention [24]. 

CVD risk related factors such as blood pressure and body mass index (BMI), Glycosylated HbA1c, HDL, LDL, and total cholesterol were also collected. BMI was computed based on the formula: 703 × body weight (lb)/[height (in) × height (in)], and the variables were measured using a calibrated scale and a stadiometer. Sitting blood pressure was measured using an automated blood pressure monitor (Omron HEM-907; Vernon Hills, IL, USA) after the participant sat quietly for about 5 min. Capillary blood samples obtained with the finger stick method were collected for the measurement of HbA1c, total cholesterol, LDL, and HDL using point-of-care testing kits. HbA1c level was measured using the A1C Now Test System (PTS Diagnostics, Whitestown, IN, USA) and total cholesterol and lipoprotein levels were measured using the Cholestech LDX System (Alere [now Abbott], San Diego, CA, USA). These data were recorded in an Excel sheet on a desktop computer. 

### 2.4. Analysis Method

Race was recoded as white or minority. One participant self-identified as Native American and another identified himself as other. Each item of the Starting the Conversation instrument was rated on a 3-point scale (0, 1, and 2) [22]. Three items were related to the consumption of healthful food, and responses were recoded to attain consistency in the creation of the composite scale score. After recoding, a summary score ranging from 0 to 16 was calculated by summing the points of all items, with higher scores indicating unhealthful dietary behaviors [22]. Smoking habit was recoded as non-smoker (never smoker or former smoker) versus current smoker. Usual living/sleeping place during homelessness was recoded as sheltered versus unsheltered (i.e., locations not designed for or ordinarily used as a regular sleeping accommodation for human beings, such as cars, parks, public spaces, bus or train stations, or similar settings).

To facilitate data analysis and interpretation, BMI, systolic and diastolic blood pressure, and each of the four CVD risk biomarkers (HbA1c, total cholesterol, LDL, and HDL) were categorized according to the clinically relevant cut-points (see Table 1 for the categorization) [25]. For example, a new variable BMI category was created in which underweight was defined as a BMI < 18.5, normal weight as a BMI between 18.5 and 24.9, overweight as a BMI between 25 to 29.9, and obese as a BMI ≥ 30 [25]. The categorization of hypertension risk followed the guidelines of the American College of Cardiology and the American Heart Association [26] and was constructed based on different combinations of systolic and diastolic blood pressure levels. Four categories of blood pressure were formed after categorization: Normal = 0, Elevated = 1, Stage 1 hypertension = 2, and Stage 2 hypertension = 3. 

### 2.5. Data Analysis 

Multivariable linear regression modeling was used to identify variables associated with each of the four CVD risk biomarkers (i.e., HbA1c, total cholesterol, LDL, and HDL). For the preliminary analysis related to the multivariable linear regression modeling, explanatory variables were initially screened for consideration in the model using bivariable associations between each explanatory variable and the four response variables (i.e., the CVD risk biomarkers). Potential explanatory variables included sociodemographic characteristics, homelessness-associated factors (frequency and duration of homelessness), and other potential risk factors related to CVD, such as blood pressure, smoking status, and BMI. 

As there were four response variables for the analysis, we fit a multivariable linear regression model for each response variable. Explanatory variables were considered as candidates for inclusion in the multivariable linear regression analysis if there was an association with the response variable (*p*-value < 0.1) in the bivariable analysis [27]. Explanatory variables with regression coefficients that had *p*-values less than 0.05 were retained in the multivariable linear regression model. All data analysis was conducted using the IBM Statistics Package for Social Sciences (SPSS) for Windows, version 28. 

Before performing the multivariable linear regression analysis, both graphical (histogram and Q-Q plot) and statistical (Shapiro-Wilk tests, skewness, and kurtosis) methods were conducted on the scores of each of the four response variables to evaluate whether the scores met the assumptions of normality. Results revealed HbA1c and HDL scores were not normally distributed, with *p*-values < 0.05 on the Shapiro–Wilk test. A log transformation was conducted on these two variables. After the log transformation, the HDL scores met the assumptions of normality (based on the Shapiro–Wilk test, *p* = 0.78). The Pearson correlation coefficient between the HDL scores and the log-transformed HDL scores was 0.98. However, HbA1c scores did not meet the normality assumption after the log, square-root, or reciprocal transformation. We then used ultra-fine transformation–a parametric power transformation [28]. After ultra-fine transformation, the HbA1c scores met the assumptions of normality (based on the Shapiro–Wilk test, *p* = 0.97). The Pearson correlation coefficient between the HDL scores and the log-transformed HDL scores was 0.77.

### 2.6. Sample Size Estimation

An online sample size calculator for multiple regression was used to estimate the minimum required number of participants to produce adequate power for finding significant relationships between the response and explanatory variables in this study [29]. Assuming the anticipated effect size (f^2^) was 0.25 (medium to large) and the inclusion of five explanatory variables, it was estimated that the study required at least 57 participants to achieve a desired statistical power level of 0.8 to provide an unbiased estimate of the parameters of the relationship between response variables and explanatory variables in the multivariable regression model at the 0.05 alpha level.

### 2.7. Sample Representation

We compared four variables (age, smoking habit, BMI category, and blood pressure) and four biomarkers of CVD risk (HbA1c, total cholesterol, LDL, and HDL) by race between the present cohort (*n* = 61) and a large cohort from our previous study (*n* = 551) [20]. Our previous cohort was recruited from the same city (Birmingham, AL) as the current study, and accrued participants over 4 years between March 2016 and March 2020 [20]. The racial composition of our previous cohort of homeless men was 377 (68.4%) Black and 174 (31.6%) white [20]. Of the eight comparisons (age, smoking habit, BMI, blood pressure, HbA1c, total cholesterol, LDL, and HDL) for each race between the participants in each study, only the BMI category of Black participants showed a significant difference at the 0.05 alpha level. A larger proportion of homeless Black individuals in the present cohort were obese (50%) compared to the previous cohort (28.3%). Results suggested that the present sample was a good representation of the homeless population in Birmingham, AL.

## 3. Results

Table 1 displays the characteristics of the 61 homeless participants enrolled in this study. The mean age of the participants was 47 years, 82% were men, and 49.2% were Black. The prevalence of smoking (85.2%) and hypertension (67.2%) in this sample was much higher than that of the general population, which is consistent with previous reports [2,12].

### 3.1. Factors Associated with Homeless Participants’ Total Cholesterol Levels

For total cholesterol, variables with a *p*-value < 0.1 included in the multivariable linear regression model were: duration of homelessness, disability status, BMI category, and education level. Participants with a longer duration of homelessness or absence of disability or who were overweight/obese or received less education exhibited higher total cholesterol levels. The multivariable linear regression model with the four factors produced R^2^ = 0.26, an adjusted R^2^ = 0.20; F (4, 56) = 4.79, *p* = 0.002. Two significant factors retained in the final model were duration of homelessness and disability status.

### 3.2. Factors Associated with Homeless Participants’ LDL Levels

For LDL, variables with a *p*-value < 0.1 included in the multivariable linear regression model were duration of homelessness and BMI category. Participants with a longer duration of homelessness or who were overweight/obese exhibited higher LDL levels. 

The multivariable linear regression model with the two factors produced R^2^ = 0.15, an adjusted R^2^ = 0.12; F (2, 58) = 5.2, *p* = 0.008. Two significant factors retained in the final model were duration of homelessness and BMI category.

### 3.3. Factors Associated with Homeless Participants’ HDL Levels

For HDL, variables with a *p*-value < 0.01 included in the multivariable linear regression model were gender, BMI category, and HbA1c level. Participants who were male or overweight/obese or who had high HbA1c levels exhibited lower HDL levels (undesirable). The multivariable linear regression model with the three factors produced R^2^ = 0.29, an adjusted R^2^ = 0.25; F (3, 57) = 7.67, *p* < 0.001. Two significant factors retained in the final model were gender and BMI category.

### 3.4. Factors Associated with Homeless Participants’ HbA1c Levels

For HbA1c, variables with a *p*-value < 0.1 included in the multivariable linear regression model were disability status, age, race, HDL level, and BMI category. Participants with a disability or a low HDL level (undesirable) or who were older, a minority, or overweight/obese exhibited higher HbA1c levels. The multivariable linear regression model with the five factors produced R^2^ = 0.38, adjusted R^2^ = 0.32; F (5, 55) = 6.70, and *p* < 0.001. Two significant factors retained in the final model were race and HDL level.

The residuals of the four regression models followed a normal distribution and were equally distributed (homoscedasticity). Multicollinearity was assessed using tolerance and the variance inflation factor; no multicollinearity was found in explanatory variables in any of the four models. 

The coefficient of each explanatory variable with significant effects on the participants’ risk factors for CVD of the univariate and multivariable regression models are presented in Table 2, Table 3, Table 4 and Table 5.

### 3.5. Sensitivity Analysis

The aim of this study was to investigate the relationship between a history of homelessness and CVD risk biomarkers in adults who were homeless. We compared the categories of each CVD risk biomarker (i.e., HbA1c, total cholesterol, LDL, and HDL) with the duration of homelessness using the Kruskal–Wallis H test, and the number of lifetime homeless episodes and usual sleeping place during homelessness (i.e., unsheltered status) using the Chi-square test. The Kruskal–Wallis H test showed that the HbA1c category was significantly related to the duration of homelessness, H(2) = 8.59, *p* = 0.014. Homeless individuals whose HbA1c levels were in the category of diabetes mellitus (median = 2) had a longer duration of homelessness than homeless individuals whose HbA1c levels were in the category of normal (median = 0.5) or prediabetes (median = 0.45). Post-hoc Mann–Whitney tests using a Bonferroni-adjusted alpha level of 0.017 (0.05/3) were used to compare all pairs of the three categories (i.e., normal, prediabetes, and diabetes). In individuals who were homeless, HbA1c levels that were in the category of diabetes were related to a longer duration of homelessness compared to HbA1c levels that were in the category of normal (z = −2.87, *p* = 0.004) or prediabetes (z = −2.57, *p* = 0.01). No significant difference in the duration of homelessness between the normal and prediabetes groups was observed. No other category of CVD risk biomarkers used in this study (i.e., total cholesterol, LDL, and HDL) was significantly related to the duration of homelessness, and no CVD risk biomarkers were significantly related to either the number of lifetime homeless episodes or unsheltered status, at the 0.05 alpha level.

## 4. Discussion

The present pilot study is the first study to report how a history of homelessness impacts CVD risk factors. Overall, the duration of homelessness showed significant positive correlations with several modifiable risk factors of CVD (diabetes mellitus, high total cholesterol, and high LDL). A longer duration of homelessness was associated with an increased risk of CVD. This is not unexpected given that previous reports have found that poor living environments contribute to high rates of CVD [30,31]. However, the frequency of homelessness or unsheltered status was not associated with any common CVD risk factors, even though duration of homelessness was significantly related to frequency of homelessness (rho = 0.29, *p* = 0.022) and unsheltered status (r_rb_ = 0.34, *p* = 0.022). More research is warranted to confirm and further explain these findings. 

As we expected in this group of homeless participants, increased BMI category showed a direct relationship with higher LDL and an inverse correlation with HDL. However, no significant association was observed between BMI category and total cholesterol levels. It has been documented that the correlation of BMI with total cholesterol level is age-dependent [32]. Older adults, especially men older than 30 years and women older than 40 years, have much weaker correlations between BMI and total cholesterol level than younger adults. The median age of these homeless participants was 46 years, with an interquartile range of 38 to 56 years, and 82% of them were men. The age factor may reduce the strength of the relationship between BMI category and total cholesterol level. 

Several homelessness-related factors with known associations with CVD risks, including dietary nutrients, alcohol consumption, and illicit substance use, were not included in the models. A large portion of homeless participants came to shelters to attend pretreatment and treatment programs for substance abuse. The shelters provide meals, therapies, and structurally supportive environments to all individuals who are homeless who seek these services. Considering the nutritious food and regular eating schedules and the treatment for drug and alcohol misuse provided by the shelters, the correlation between these homelessness-related factors and CVD risk factors may be attenuated in this group of individuals who were homeless. 

There was no significant association between the three categories (desirable, borderline, and high) of total cholesterol or LDL levels and the duration of homelessness, and we attribute this mainly to the extremely small sample (*n* ≤ 3) in the category of “high” for both biomarkers. The prevalence of smoking and hypertension in the homeless population was notably high. However, no significant association was identified between these two CVD risk factors and the duration of homelessness or CVD risk biomarkers, which may partly be due to the small proportions and the small numbers of non-smokers (<15%, *n* = 9) or participants with normal blood pressure (<25%, *n* = 15) in this sample. In particular, extra adiposity has been shown to counterbalance the consequences of smoking to lipoprotein levels [33]. The mean BMIs of both male and female participants in this study was 30.4 kg/m^2^, whereas in the Framingham Study the mean BMI was only 24.8 kg/m^2^ in women and 26.8 kg/m^2^ in men [34]. The high BMI values in the current study may partially obscure correlations between smoking and these CVD risk factors.

While this study provides important insights into the relationship between duration of homelessness and CVD risk factors, its results should be considered in light of several limitations. The most important limitations are the convenience sampling from only two homeless shelters in one city and the small size of the sample population, which limit the generalizability of the results to homeless populations in other cities or states. A larger sample size is needed to explore other homelessness-related factors. A further limitation is the limited accuracy and precision of the self-reported nature of homelessness. As the mobility of this population is extremely high, incorporating quantifiable information–such as attendance at a shelter or soup kitchen–to verify individuals’ duration of homelessness may provide objective data, though collecting this data presents challenges. In addition, about 10% of participants included in the data analysis had heart-related problems, which may skew the predictive sensitivity of the measures. Finally, it is not always feasible to obtain the fasting blood samples that are required for measurement of triglyceride and glucose levels in individuals who are homeless. Therefore, triglyceride level was not included, and HbA1c value rather than glucose level was used in the data analyses.

People who are homeless experience stark disparities in the incidence and outcomes of CVD compared to those with permanent housing [1,2]. It is necessary to understand the impacts of a history of homelessness on CVD risk factors to address the issues involved in these health disparities effectively. In addition, multidisciplinary collaboration among housing authorities, health services, and vocational specialists are required–not only to reduce the duration of homelessness, but to prevent it. 

## 5. Conclusions

The findings from this study suggest that an increased duration of homelessness was positively associated with particular CVD risk factors, including increased total cholesterol and LDL and diabetes mellitus. Our findings have important implications for both the health of individuals and for efforts to reduce health disparities in the homeless population. Specifically, given the multifactorial challenges faced by people who are homeless, housing policies and employment services that help these individuals achieve timely re-entry into the job market and secure permanent living places may serve to protect their health. 

## Figures and Tables

**Table 1 ijerph-19-14698-t001:** Characteristics of homeless participants (*n* = 61).

Variable	Number (%)/Mean ± SD (Range)
Age (years)	47.1 ± 12.5 (20–75)
Gender	
male	50 (82%)
female	11 (18%)
Race	
white	29 (47.5%)
minority (30 Black persons)	32 (52.5%)
Marital status	
married	2 (3.3%)
widowed	3 (4.9%)
separated/divorced	28 (45.9%)
never married	28 (45.9%)
Veteran	6 (9.8%)
LGBTQ	4 (6.6%)
Education level	
less than high school	15 (24.6%)
high school graduate	29 (47.5%)
associate degree	14 (23.0%)
bachelor’s degree	3 (4.9%)
Employed	5 (8.2%)
Had adverse childhood experience	27 (44.3%)
Disability status	17 (27.9%)
Smoking habit	
never smoker	7 (11.5%)
former smoker	2 (3.3%)
someday smoker	0 (0.0%)
every day smoker	52 (85.2%)
Duration of homelessness (years)	1.3 ± 1.7 (0.1 to 7.5 years)
Live in a sheltered place	41 (67.2%)
Homeless episode	
first or 1 time	22 (36.1%)
2–5 times	30 (49.2%)
>5 times	9 (14.8%)
BMI category	
underweight	2 (3.3%)
normal weight	15 (24.6%)
overweight	15 (24.6%)
obese	29 (47.5%)
Blood pressure category	
normal	15 (24.6%)
elevated	5 (8.2%)
stage 1 hypertension	20 (32.8%)
stage 2 hypertension	21 (34.4%)
Glycosylated HbA1c	
normal (<5.7%)	41 (67.2%)
prediabetes (5.7–6.4%)	14 (23.0%)
diabetes (≥6.5%)	6 (9.8%)
Total cholesterol	
desirable (<200 mg/dL)	44 (72.1%)
borderline (200–239 mg/dL)	14 (23.0%)
high/undesirable (≥240 mg/dL)	3 (4.9%)
LDL	
desirable (<130 mg/dL)	50 (82%)
borderline (130–159 mg/dL)	9 (14.8%)
high/undesirable (≥160 mg/dL)	2 (3.3%)
HDL	
desirable (≥60 mg/dL)	11 (18.0%)
borderline (40–59 mg/dL)	22 (36.1%)
low/undesirable (<40 mg/dL)	28 (45.9%)

Abbreviations: BMI, body mass index; HbA1c, hemoglobin A1c; HDL, high-density lipoprotein; LDL, low-density lipoprotein.

**Table 2 ijerph-19-14698-t002:** Bivariable and multivariable linear regression analyses examining factors associated with HbA1c level (*n* = 61).

Explanatory Variable	Bivariable Analysis			Multivariable Analysis		
	Estimate (Beta)	Standard Error (SE)	*p*-Value	Estimate (Beta)	Standard Error (SE)	*p*-Value
Race (white = 1, Black = 2)	4.74	1.15	<0.001 ***	3.83	1.22	0.003 ***
HDL level	−0.08	0.04	0.060	−0.08	0.04	0.037 *
BMI	0.16	0.09	0.064	0.08	0.08	0.332
Age	0.118	0.05	0.022 *	0.07	0.05	0.172
Disability status	3.90	1.36	0.006 **	1.98	1.26	0.123

Note: *** *p* < 0.001, ** *p* < 0.01, and * *p* < 0.05. Abbreviations: BMI, body mass index; HDL, high-density lipoprotein.

**Table 3 ijerph-19-14698-t003:** Bivariable and multivariable linear regression analyses examining factors associated with total cholesterol level (*n* = 61).

Explanatory Variable	Bivariable Analysis			Multivariable Analysis		
	Estimate (Beta)	Standard Error (SE)	*p*-Value	Estimate (Beta)	Standard Error (SE)	*p*-Value
Duration of homelessness	8.28	3.01	0.008 **	7.02	2.92	0.020 *
Disability status	−23.16	11.31	0.045 *	−24.24	10.48	0.024 *
Education level	−10.68	6.28	0.094	−8.68	5.91	0.148
BMI category	9.83	5.63	0.086	10.25	5.28	0.057

Note: ** *p* < 0.01, and * *p* < 0.05. Abbreviation: BMI, body mass index.

**Table 4 ijerph-19-14698-t004:** Bivariable and multivariable linear regression analyses examining factors associated with LDL level (*n* = 61).

Explanatory Variable	Bivariable Analysis			Multivariable Analysis		
	Estimate (Beta)	Standard Error (SE)	*p*-Value	Estimate (Beta)	Standard Error (SE)	*p*-Value
Duration of homelessness	5.95	2.57	0.024 *	5.13	2.52	0.046 *
BMI category	11.27	4.62	0.018 *	9.89	4.55	0.034 *

Note: * *p* < 0.05. Abbreviation: BMI, body mass index.

**Table 5 ijerph-19-14698-t005:** Bivariable and multivariable linear regression analyses examining factors associated with HDL level (*n* = 61).

Explanatory Variable	Bivariable Analysis			Multivariable Analysis		
	Estimate (Beta)	Standard Error (SE)	*p*-Value	Estimate (Beta)	Standard Error (SE)	*p*-Value
Gender (male = 1, female = 2)	0.16	0.05	0.002 **	0.13	0.05	0.006 **
BMI category	−0.07	00.02	0.002 **	−0.06	0.02	0.005 **
HbA1c level	−0.03	0.01	0.08	−0.02	0.01	0.204

Note: ** *p* < 0.01. Abbreviations: BMI, body mass index; HbA1c, hemoglobin A1c.

## Data Availability

Not applicable.

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
