# Peer review of "Associations between Duration of Homelessness and Cardiovascular Risk Factors: A Pilot Study"

_ijerph, 2022, doi:10.3390/ijerph192214698_

Round 1
Reviewer 1 Report (Previous Reviewer 3)
The authors have addressed this reviewer's comments/concerns in their point by point response and revised manuscript.
Reviewer 2 Report (Previous Reviewer 2)
I liked the great work of the authors in improving the manuscript in response to my comments and the comments of the second reviewer. I have no other questions.
This manuscript is a resubmission of an earlier submission. The following is a list of the peer review reports and author responses from that submission.
Round 1
Author Response
We thank the reviewer for the positive remarks and comments on our manuscript. We have responded to the reviewer’ specific critiques and incorporated the suggestions. The subsequent changes have significantly strengthened the manuscript.
- L 61 Please include number of eligible participants and calculate participation rate.
We appreciate this suggestion, and have provided the overall number eligible participants and an enrollment rate.
2. How were the 2 participating shelters selected?
The authors inquired about study participation at more than 10 homeless shelters in Birmingham, AL, and only two shelters agreed to participate. As this was a convenience sample, future studies will seek to expand the number of participants.
3. L102 Include method of data recording, e.g., pencil-and-paper, computer.
Thank you for the suggestion. To address this, we added a sentence: “…investigators interviewed participants and recorded their responses in an online questionnaire created with QualtricsXM …”
4. Relocate description of method of scoring for each instrument (including STC, BMI and BP) to section, “Description of Instruments” (or analysis method).
We appreciate this suggestion. This description has been relocated.
5. L169 Readers are likely to be unfamiliar with Mallows' Cp value; a brief explanation is called for.
Thank you for the suggestion. The data have been analyzed, and the Mallows’ Cp value was removed from the manuscript.
6. A graphic image (model) displaying the significant relationships would concisely show findings and enhance reader understanding.
Thank you for the suggestion. We would like to include a graphic displaying the significant relationships, but the interactions between the homelessness factors and the risk biomarkers of CVD are complex. The preliminary data in this pilot study may not reveal the full complexity of all the interactions and relationships. We are concerned that a graphic image oversimplifies the interactions and relationships, and may mislead readers. The significant relationships have been summarized concisely in the conclusion.
7. L 270 Include in limitations the impact of the convenience sample on study findings. Cross sectional study, only from two shelters
Thank you for the suggestion. The limitation regarding the impact of the convenience sample on study findings was included in the last paragraph of the discussion section.
8. Please comment on the authors' hypotheses regarding the impact of duration (vs. frequency) of homelessness on CVD risk factors?
Thank you for the suggestion. The current study focuses on exploring the correlations between the duration/frequency of homelessness and CVD risk factors and biomarkers (total cholesterol, LDL, HDL, HbA1c) in homeless adults, and the findings from this study suggest that an increased duration of homelessness was positively associated with several CVD risk factors.
9. Consider including a marker (e.g., symbol) on tables, distinguishing significant p
Thank you for the suggestion. We have distinguished significant p values in tables with the following symbols: *** (p < .001), ** (p < 0.01), and * (p < 0.05).
Reviewer 2 Report
I would like to thank the authors for the opportunity to review their manuscript on the association between duration of homelessness and cardiovascular risk factors. In their manuscript, the authors presented some new scientific facts, showed that the duration of homelessness is associated with increased levels of glycated hemoglobin, total cholesterol and low-density lipoprotein cholesterol. However, when reviewing the manuscript, I had a number of comments.
Major:
1. First of all, it is not clear why the authors analyzed only four of the known modifiable risk factors (the level of glycated hemoglobin, total cholesterol, high- and low-density lipoprotein cholesterol). Why didn't they include in the analysis, for example, blood pressure and smoking status? It may be recalled that in the article by Szerlip MI and Szerlip HM (2002) it was shown that smoking and hypertension are significantly more prevalent in the homeless population than in a matched cohort (in contrast to high cholesterol and diabetes). For some reason, the authors ignored these results, I would like to see their explanation of this approach.
2. The term of the authors "chronic medical status" is completely incomprehensible. What diseases are meant by this term? Does arterial hypertension or diabetes mellitus fall under this section? Or did the homeless already have some kind of cardiovascular disease? And then how correct is it to assess the presence of risk factors for an already existing pathology? An explanation is required from the authors. It would be best to give specific nosological forms that the homeless suffer from.
3. The statistical analysis section is not well written. First, there is no such subheading in principle, which makes it difficult to understand the statistical methods used by the authors. Secondly, there is no information about checking the data for normal distribution. If the distribution differs from normal, then multiple linear regression analysis is incorrect. There is also no information about the data presentation format, it differs in the cases of normal and non-parametric distribution. In addition, when conducting multiple linear regression analysis, quantitative data with a normal distribution are used not only in the dependent, but also in the explanatory variable. Nevertheless, the authors used ordinal values ​​as explanatory variable in a number of models, which can distort the results of the analysis.
Minor:
1. The phrase "Biomarkers related to CVD risk factors, i.e., blood pressure, body mass index (BMI), hemoglobin A1c (HbA1c), HDL, LDL, and total cholesterol, were also collected" is unfortunate. The level of blood pressure and body mass index are difficult to attribute to biomarkers, which usually include biochemical or other laboratory parameters, and not physiological constants of the body.
2. It seems to me that the line "Duration of homelessness (yr) 13 ± 1.7 (0.1 to 7.5 yr)" in table 1 contains an error. Check, please.
Reference:
Szerlip MI, Szerlip HM. Identification of cardiovascular risk factors in homeless adults. Am J Med Sci. 2002 Nov;324(5):243-6. doi: 10.1097/00000441-200211000-00002.
Author Response
We thank the reviewer for the positive remarks and comments on our manuscript. We have responded to the reviewer’ specific critiques and incorporated the suggestions. The subsequent changes have significantly strengthened the manuscript.
Major
- First of all, it is not clear why the authors analyzed only four of the known modifiable risk factors (the level of glycated hemoglobin, total cholesterol, high- and low-density lipoprotein cholesterol). Why didn't they include in the analysis, for example, blood pressure and smoking status? It may be recalled that in the article by Szerlip MI and Szerlip HM (2002) it was shown that smoking and hypertension are significantly more prevalent in the homeless population than in a matched cohort (in contrast to high cholesterol and diabetes). For some reason, the authors ignored these results, I would like to see their explanation of this approach.
Thank you for this comment. It is true that both smoking and hypertension are prevalent in the homeless population. For example, the prevalent rate of smoking was 85.2% and hypertension was 67.2% in our sample. We reported incidence of smoking and hypertension in the current sample in Table 1. While we are aware of the negative health consequences of hypertension and smoking, these factors were not significantly related to the CVD (cardiovascular disease) risk biomarkers in our data analysis. Nonetheless, the reviewer is correct, and we have added these data to our results, as well as a discussion of the possible reasons for the lack of a significant relationship.
- The term of the authors "chronic medical status" is completely incomprehensible. What diseases are meant by this term? Does arterial hypertension or diabetes mellitus fall under this section? Or did the homeless already have some kind of cardiovascular disease? And then how correct is it to assess the presence of risk factors for an already existing pathology? An explanation is required from the authors. It would be best to give specific nosological forms that the homeless suffer from.
Thank you for this comment. The variable should be “disability status,” a self-report item, and we have corrected this mistake.
- The statistical analysis section is not well written. First, there is no such subheading in principle, which makes it difficult to understand the statistical methods used by the authors. Secondly, there is no information about checking the data for normal distribution. If the distribution differs from normal, then multiple linear regression analysis is incorrect. There is also no information about the data presentation format, it differs in the cases of normal and non-parametric distribution. In addition, when conducting multiple linear regression analysis, quantitative data with a normal distribution are used not only in the dependent, but also in the explanatory variable. Nevertheless, the authors used ordinal values ​​as explanatory variable in a number of models, which can distort the results of the analysis.
We appreciate the reviewer’s suggestions and have clarified our analyses. We have provided summary data for each measure and a test of normality.
Minor:
- The phrase "Biomarkers related to CVD risk factors, i.e., blood pressure, body mass index (BMI), hemoglobin A1c (HbA1c), HDL, LDL, and total cholesterol, were also collected" is unfortunate. The level of blood pressure and body mass index are difficult to attribute to biomarkers, which usually include biochemical or other laboratory parameters, and not physiological constants of the body.
Thank you for the correction. The sentence was revised as “CVD risk factors such as blood pressure and body mass index (BMI) and CVD risk biomarkers, i.e., HbA1c, HDL, LDL, and total cholesterol, were also collected.”
- It seems to me that the line "Duration of homelessness (yr) 13 ± 1.7 (0.1 to 7.5 yr)" in table 1 contains an error. Check, please.
Thank you for the correction. There was a typo in the line. It is corrected as “Duration of homelessness (yr) 1.3 ± 1.7 (0.1 to 7.5 yr).”
Reviewer 3 Report
This is a well designed study to assess associations of duration of homelessness and four important CVD risk factors. The subject of the manuscript has a high clinical significance and the manuscript is well written.
Major comments:
1. In the absence of data from control group or other appropriate comparison group the authors should provide average+/- SD values of all 4 risk factors studied in a separate table and compare these values with values available in the literature for general populations with similar age and gender. This will allow readers to assess if levels of these risk factors increased (except HDL-C for which levels should go down) compared to non-CVD group. In the future studies they should include a control group.
2. If possible authors should assess if factors such as divorce, financial hardship or psychological factors are playing a role in high risk of CVD in this group (i.e., factors that might have caused homelessness in the first place).
Minor Comments:
1. In Table 1 is data presented for Duration of homelessness correct? Are the data in the parenthesis represent range and if so how it is less than average?
2. Reference 19: There is no volume and page numbers provided.
Author Response
We thank the reviewer for the positive remarks and comments on our manuscript. We have responded to the reviewer’ specific critiques and incorporated the suggestions. The subsequent changes have significantly strengthened the manuscript.
Major comments:
- In the absence of data from control group or other appropriate comparison group the authors should provide average+/- SD values of all 4 risk factors studied in a separate table and compare these values with values available in the literature for general populations with similar age and gender. This will allow readers to assess if levels of these risk factors increased (except HDL-C for which levels should go down) compared to non-CVD group. In the future studies they should include a control group.
Thank you for the suggestion. We would like to compare the CVD and non-CVD groups in future studies. The current study focuses on exploring the correlations between the duration and frequency of homelessness and CVD risk biomarkers (total cholesterol, LDL, HDL, and HbA1c) in homeless adults; 90% of the participants in this study had no CVD.
- If possible authors should assess if factors such as divorce, financial hardship or psychological factors are playing a role in high risk of CVD in this group (i.e., factors that might have caused homelessness in the first place).
Thank you for the suggestion. We are aware of the importance of assessing factors such as divorce, financial hardship, and psychological conditions that may contribute to the high risk of CVD among people who are homeless before they become unhoused. However, the current study focuses on the correlations between the duration and frequency of homelessness and CVD risk biomarkers (total cholesterol, LDL, HDL, and HbA1c) in homeless adults. Future work should include these factors, which may cause homelessness and contribute to CVD.
Minor Comments:
- In Table 1 is data presented for Duration of homelessness correct? Are the data in the parenthesis represent range and if so how it is less than average?
Thank you for the correction. There was a typo in the line. It is corrected as “Duration of homelessness (yr) 1.3 ± 1.7 (0.1 to 7.5 yr).”
- Reference 19: There is no volume and page numbers provided.
Thank you for the correction. The volume and page numbers were added to the reference.